# Identification of Biomarkers Related to M2 Macrophage Infiltration in Alzheimer’s Disease

**DOI:** 10.3390/cells11152365

**Published:** 2022-08-01

**Authors:** Caixiu Lin, Congcong Xu, Yongji Zhou, Anqi Chen, Baiye Jin

**Affiliations:** 1The First Affiliated Hospital, Zhejiang University School of Medicine, Hangzhou 310009, China; lincaixiu123@zju.edu.cn (C.L.); 21618215@zju.edu.cn (C.X.); 2Department of Neurology, Affiliated Hangzhou First People’s Hospital, Zhejiang University School of Medicine, Hangzhou 310009, China; zjy0423@zju.edu.cn; 3The Research Institute of Advanced Technologies, Ningbo University, Ningbo 315211, China; chenanqi@nbu.edu.cn

**Keywords:** Alzheimer’s disease, M2 macrophages, toll-like receptor 2, immune infiltration

## Abstract

Many studies have demonstrated that neuroinflammation contributes to the onset and development of Alzheimer’s disease (AD). The infiltration of immune cells in the brain was observed in AD. The purpose of the present study was to verify potential mechanisms and screen out biomarkers related to immune infiltration in AD. We collected the expression profiling datasets of AD patients and healthy donors from the NCBI’s Gene Expression Omnibus (GEO) database. We confirmed that immune-related mechanisms were involved in AD using differentially expressed genes analysis and functional enrichment analysis. We then found that M2 macrophage infiltration was most positively correlated with AD according to the CIBERSORT algorithm and a weighted gene co-expression network analysis (WGCNA). *TLR2*, *FCGR2A*, *ITGB2*, *NCKAP1L* and *CYBA* were identified as hub genes correlated with M2 macrophage infiltration in AD. Furthermore, the expression levels of these hub genes were positively correlated with Aβ42 and β-secretase activity. A diagnostic model of these hub genes was constructed, which showed a high area under the curve (AUC) value in both the derivation and validation cohorts. Overall, our work further expanded our understanding of the immunological mechanisms of AD and provided new insights into therapeutic strategies in AD.

## 1. Introduction

Alzheimer’s disease (AD) is the most common neurodegenerative disorder, characterized by progressive and irreversible memory loss and cognitive impairment. It was estimated that AD affected more than 50 million people worldwide in 2018 [1]. It has been predicted that this number will triple by 2050, which will confer a heavy economic burden on society [2]. Extracellular amyloid plaques and intracellular neurofibrillary tangles (NFTs) are typical pathological hallmarks in AD, and these remain the gold standard for diagnosis. Currently, AD is recognized as a continuum with the notion that AD starts 20 years or more before symptoms appear and progresses gradually during aging [3,4]. At present, there are no effective therapeutic strategies to stop disease progression. Therefore, improved understanding of the pathophysiology of AD is required for the diagnosis and treatment of AD.

In the past few decades, several theories have been presented to explain the pathogenesis of AD. Neuroinflammation is considered to be one of the important contributors to AD pathology [5,6]. At the beginning of AD, an acute inflammatory response is activated by emerging neurotoxic amyloid-β (Aβ) peptides and NFTs to aid pathogen or toxin clearance and restore tissue homoeostasis. Yet, sustained exposure and immune activation cause chronic neuroinflammation, ultimately leading to enlarged pathological changes and neuronal damage [7,8]. The central nervous system (CNS) is traditionally considered to be an immune-privileged site because of the presence of the blood–brain barrier (BBB). The main players involved in this neuroinflammation are primarily microglia (resident macrophages) and astroglia. However, a growing body of evidence argues that peripheral immune cells are able to infiltrate into the brain and exert either detrimental or beneficial effects in AD. For example, massive bone-marrow-derived monocytes are recruited and appear within the core of amyloid plaques, with high efficiency in eliminating amyloid deposits [9]. Neutrophils could adhere to and migrate inside brain vessels in individuals with AD [10]. The neutrophil depletion or blockage of neutrophil trafficking was found to reduce AD-like neuropathology and ameliorate cognitive dysfunction in 3xTg-AD mice [10]. CD8+ T cells have also been identified in human postmortem hippocampi from individuals with AD, which might directly participate in regulating synaptic plasticity and lead to neuronal dysfunction [11]. These studies shed light on the communication between peripherally derived immune cells and the brain in AD, but the mechanisms that mediate immune infiltration underlying AD have not been clarified. The exploration of the changes in immune cells’ composition and the gene expression levels of immune cells in AD will facilitate a better understanding of the details of disease progression.

The aim of this study was to explore and identify the key pathways and potential genes related to AD immune cells’ infiltration. The CIBERSORT algorithm was used to calculate the composition of immune cells using gene expression profiles of human prefrontal cortexes. Then, weighted gene co-expression network analysis (WGCNA) was conducted to identify the most important module so as to identify hub genes related to immune infiltration levels. Five hub genes were screened out and then validated using other GEO expression profiles of AD. Additionally, we analyzed the association between the expression levels of hub genes and Aβ42. Our results showed that these hub genes are potential biomarkers for disease progression and are mechanically involved in M2 macrophage infiltration in AD.

## 2. Materials and Methods

### 2.1. Data Source

We searched the NCBI Gene Expression Omnibus (GEO) database (https://www.ncbi.nlm.nih.gov/geo/, accessed on 26 April 2022) to collect datasets that included gene expression profiles of individuals with AD and healthy controls (HCs). GSE33000 contained the gene expression profiles of prefrontal cortexes from 310 postmortem samples of individuals with AD and 157 samples from HCs without dementia, which were used as matched clinical data. GSE44770 and GSE118553 were used for independent external validation. GSE44770 consisted of 230 prefrontal cortex expression profiles from 129 AD patients and 101 HCs without dementia. From GSE118553, we selected the expression profiles of prefrontal cortexes derived from 102 postmortem brains, among which 33 subjects were diagnosed with AD and 33 had asymptomatic AD (AsymAD). GSE106241 was downloaded to analyze the relationship between the hub genes and Aβ42.

### 2.2. Differentially Expressed Genes and Enrichment Analysis

The limma package from RStudio software was used to infilter differentially expressed genes (DEGs) between AD and HCs with adjusted *p* < 0.05 and log2 fold change >1 or <−1 in GSE33000 datasets. Gene Ontology (GO) functional annotation and Kyoto Encyclopedia of Genes and Genomes (KEGG) pathway analysis of DEGs were conducted by using the R package ‘clusterProfiler’. Pathways enriched with adjusted *p* < 0.05 were regarded as statistically significant. The visualization of GO and KEGG term analysis was achieved using GOPlot (version 1.0.2; authors: Walter Wencke, Fátima Sánchez-Cabo and Mercedes Ricote; Madrid, Spain).

### 2.3. Evaluation of Immune Infiltration

The CIBERSORT algorithm is widely used in the calculation of the abundance of immune cells in the microenvironment. LM22 is a leukocyte gene signature matrix, which can distinguish 22 human hematopoietic cell phenotypes, including seven T cell types, naïve and memory B cells, plasma cells, natural killer (NK) cells and myeloid subsets [12]. In this study, to identify the infiltration level of peripheral immune cells to the brain, we utilized CIBERSORT-based deconvolution combined with LM22 to measure the relative proportion of 22 types of immune subpopulations in prefrontal cortex samples from GSE33000.

### 2.4. Weighted Gene Co-Expression Network Analysis

A total of 19,523 genes from the GSE33000 datasets were extracted for WGCNA using the R package “WGCNA”. Firstly, a matrix of similarity was generated by calculating the Pearson’s correlation value between all the gene pairs. Next, we converted the similarity matrix into the adjacency matrix using a power adjacency function with amn = |cmn| β (amn = adjacency between paired genes; cmn = Pearson’s correlation between paired genes). Then, the adjacency matrix was transformed into the topological overlap matrix at the soft power threshold of 6. Finally, clusters of genes with similar patterns were categorized into different modules using a dynamic hybrid tree-cutting algorithm (minimum module size cut-off = 60).

### 2.5. Identification of Hub Genes

Candidate hub genes of the co-expression module were selected based on a high module membership (MM) and gene significance (GS). MM indicated the absolute value of the Pearson’s correlation between each gene and the module eigengene, while GS was measured according to the correlation between each gene and the clinical trait. Those genes with MM > 0.8 and GS > 0.45 were judged as candidate hub genes. At the same time, we generated a protein–protein interaction (PPI) network based on all the genes in the hub module to identify central genes using the Search Tool for the Retrieval of Interacting Genes (STRING; https://string-db.org/, accessed on 14 May 2022) database. We defined genes with weight >0.15 and node connectivity >15 as hub nodes and visualized the result using Cytoscape 3.9.1 (https://cytoscape.org/, accessed on 14 May 2022). Using a publicly available online tool, Venn diagrams were plotted based on overlapping analysis between hub nodes and the candidate hub genes (http://bioinformatics.psb.ugent.be/webtools/Venn/, accessed on 15 May 2022).

### 2.6. Validation of Hub Genes

We employed external validation datasets (GSE44770, GSE118553 and GSE106241) to verify these hub genes. The Wilcoxon signed-rank test was applied to compare the differences between individuals with AD and HCs. The diagnostic performance of the hub genes was assessed using ROC analysis. The correlations between hub genes and other indexes were assessed using Spearman’s or Pearson’s correlation test.

## 3. Results

### 3.1. Identification of DEGs and Enrichment Analysis in Individuals with AD and HCs

The research flowchart of this work is displayed in Figure 1. We downloaded GSE33000 datasets from the GEO database. A total of 467 samples (357 from individuals with AD and 110 from HCs) were available for DEGs analysis. A total of 222 genes were significantly upregulated, while 229 genes were downregulated in AD patients (Figure 2A,B, Appendix A). Enriched terms were sorted using score-based ranking to generate the top enriched GO and KEGG terms (Figure 2C,D). Several immune- and inflammation-related pathways (neutrophil activation (GO:0042119); the regulation of inflammatory response (GO:0050727); neutrophil activation involved in immune response (GO:0002283); acute inflammatory response (GO:0002526); neutrophil-mediated immunity (GO:0002446); humoral immune response (GO:0006959); Staphylococcus aureus infection (hsa05150); phagosome (hsa04145)) were upregulated according to the z-scores (Appendix A).

### 3.2. Identification of Immune Infiltration Pattern in AD

According to the aforementioned functional enrichment analysis, we elicited that immune-related pathways were highly enriched in AD patients. Therefore, we calculated the fractions of 22 types of immune cells by using the R package “CIBERSORT”. The distribution of various immune cells differed between individuals, and macrophages comprised the largest population of infiltrating immune cells (Figure 3A). In addition, the fractions of M2 macrophages, CD4 memory resting T cells, resting NK cells, monocytes, neutrophils and M1 macrophages were remarkably higher in the AD group compared with the HC group, while the fractions of memory B cells, CD8 T cells, plasma cells, activated mast cells, activated NK cells and follicular helper T cells were lower in the AD group (Figure 3B,C).

### 3.3. Identification of Key Modules via WGCNA

A WGCNA was performed to screen for the key module most strongly related to the aforementioned infiltrated immune cells. After performing sample clustering to plot the sample tree, we selected the cut height of 150 and mini sample size of 10 (Appendix A). Then, a total of four samples were removed as outliers, and we built the sample dendrogram and trait heatmap (Appendix A). We constructed a scale-free co-expression network using the soft threshold power of 6 (scale free R^2^ = 0.9) (Figure 4A). According to the similarity between genes, 21 modules were identified (Figure 4B). Among the 21 modules, we found that the cyan module was most positively correlated to M2 macrophages (Cor = 0.52, *p* < 0.001) (Figure 4C).

### 3.4. Identification of Hub Genes

A total of 479 genes were included in the cyan model (Appendix A). We selected the highly connected genes in the cyan module to mine potential key molecules linked to the infiltration level of M2 macrophages. On the basis of the cut-off standard (MM > 0.8 and GS > 0.45), 49 genes were filtered out as candidate hub genes (Figure 5A, Appendix A). The interactions of genes from the cyan module were visualized in the PPI network; 15 genes with weight >0.15 and connectivity >15 (node/edge) were identified as central nodes (Figure 5B, Appendix A). We intersected 49 candidate hub genes and 15 hub nodes and finally identified five genes as hub genes (Figure 5C).

The expression values of the five genes were all positively correlated with the infiltration level of M2 macrophages (Figure 6A). Furthermore, we estimated the efficacy of these hub genes to differentiate individuals with AD and HCs. The AUC values of *TLR2, CYBA, ITGB2, FCGR2A* and *NCKAP1L* were 0.897 (95% CI, 0.863–0.932), 0.860 (95% CI, 0.822–0.898), 0.868 (95% CI, 0.832–0.904), 0.851 (95% CI, 0.812–0.889) and 0.768 (95% CI, 0.722–0.814), respectively (Figure 6B). Additionally, a logistic regression model was built based on the five hub genes, achieving an AUC of 0.909 (95% CI, 0.876–0.941) (Figure 6C).

### 3.5. Validation of Hub Gens

In the GSE44770 datasets, these hub genes (*TLR2**,** CYBA**,** ITGB2**,** FCGR2A* and *NCKAP1L*) exhibited higher expression levels in AD patients than HCs (*p* < 0.05) (Figure 7A). To validate the diagnostic significance of the hub genes in the GSE44770 dataset, ROC analysis was conducted. The AUC values of *TLR2, CYBA, FCGR2A, NCKAP1L* and *ITGB2* were 0.913 (95% CI, 0.872–0.954), 0.883 (95% CI, 0.836–0.929), 0.879 (95% CI, 0.832–0.927), 0.804 (95% CI, 0.745–0.863) and 0.864 (95% CI, 0.815–0.913), respectively. After logistic regression analysis, the AUC for the model of five genes was 0.934 (95% CI, 0.898–0.970). Finally, the positive correlation between the five hub genes and M2 macrophages’ infiltration was verified (R > 0.5).

From the expression data of frontal cortexes from GSE118553 (Figure 7B), we found that the expression levels of *TLR2*, *CYBA*, *ITGB2* and *FCGR2A* were higher in the AD group than in the HC group. No significant difference in gene expression was found between the AD group and AsymAD group, except for that of *FCGR2A*, implying the expression levels of these hub genes were probably related to the course of the disease. To further verify the diagnostic efficacy of these hub genes, ROC analysis was carried out for *TLR2*, *CYBA*, *ITGB2* and *FCGR2A*, which significantly differentiated the AD and HC groups. The AUC values of *TLR2, CYBA, ITGB2* and *FCGR2A* were 0.735 (95% CI, 0.597–0.873), 0.724 (95% CI, 0.589–0.860), 0.779 (95% CI, 0.665–0.894) and 0.721 (95% CI, 0.586–0.855), respectively. After logistic regression analyses, the AUC for the model of four genes was 0.846 (95% CI, 0.748–0.943). In this dataset, the expression levels of *CYBA, FCGR2A* and *ITGB2* were positively associated with M2 macrophages’ infiltration.

Furthermore, we analyzed the relationships between these hub genes and Aβ42 by utilizing expression profiles from GSE106241. We found that the expression levels of *TLR2, CYBA* and *FCGR2A* were positively associated with Aβ42 levels, and *TLR2, CYBA, FCGR2A* and *NCKAP1L* were positively associated with β-secretase activity (*p* < 0.05) (Figure 7C).

## 4. Discussion

Chronic neuroinflammation is considered to be a crucial feature of AD [5]. Numerous studies have described a complex picture of neuroinflammation in AD, involving multiple factors and cell types [5,13]. Microglia are known as brain-resident macrophages which continuously monitor the environment of the CNS and recognize various pathological insults, working as the first line of defense against pathogens in the CNS [14]. In AD, microglia can be triggered by soluble Aβ oligomers and Aβ fibrils, resulting in the release of proinflammatory cytokines and chemokines, hence subsequently causing neurodegeneration [15,16]. When amyloid plaque accumulates, activated microglia show the ability to enable Aβ fibrils’ uptake and the phagocytosis of Aβ, implying their potential for clearing amyloid accumulation in AD [17,18,19].

In addition to microglia, peripheral macrophages (used interchangeably with the term “bone-marrow-derived mononuclear phagocytes” in this article) are highly phagocytic cells, which can also migrate into the brain and have an increased capacity to clear Aβ fibrils in both cultured cells and animal models [20,21,22]. Simard and colleagues created a chimeric mouse model by irradiating APP/PS1 mice and transplanting bone marrow cells into their bloodstreams; they demonstrated that peripheral macrophages but not microglia were more efficient in eliminating Aβ via phagocytosis [9]. Joseph El Khoury and colleagues showed that the deletion of Ccr2, a chemokine receptor mediating the accumulation of mononuclear phagocytes at sites of inflammation, impaired the trafficking of peripheral macrophages to Aβ composition. More notably, this insulted recruitment of peripheral macrophages led to accelerated early disease progression and increased mortality [23]. An additional study provided further evidence that the enhanced migration of peripheral macrophages to the CNS, induced by glatiramer acetate immunization, curbed Aβ neuropathology and prevented cognitive decline in APP/PS1 mice [24]. These findings suggested that the recruitment of peripheral macrophages to the CNS might be a potentially therapeutic target for AD. However, the particular macrophage cell subtype was not specified. Macrophage subsets can be classified as at least two effectors responding to local microenvironment. M1 macrophages exhibit a pro-inflammatory response, while M2 macrophages are believed to be anti-inflammatory, involved in tissue repair and in terminating inflammation [25,26,27,28]. A limited study found that M2 macrophage transplantation could reduce neuronal loss, ameliorate inflammation response and improve cognitive function in AD model rats, implying a protective effect of this subtype of macrophages in AD [29]. In this work, we found that there was a remarkable difference in the proportion of immune cells between individuals with AD and HCs by utilizing CIBERSORT deconvolution analysis, and M2 macrophages’ infiltration was highly correlated to AD via WGCNA.

We identified five hub genes (*TLR2*, *CYBA*, *ITGB2*, *FCGR2A* and *NAKCPIL*) involved in the infiltration of M2 macrophages in AD through WGCNA and the PPI network. Toll-like receptor 2 (TLR2) is a member of the TLR family, which includes innate immunity sensors that are able to recognize various invading pathogen-associated molecular patterns (PAMPs) as well as endogenous damage-associated molecular patterns (DAMPs) [30,31]. TLR2 could be triggered by Aβ42, and the expression level of TLR2 is elevated in the prefrontal cortex in AD patients relative to levels in individuals with mild cognitive impairment (MCI) and HCs based on autopsy [32]. Some studies indicated a protective role of TLR2 in AD, as the activation of TLR2 on microglia promoted enhanced Aβ42 uptake [17,33]. Zhou et al. revealed that TLR2 deficiency exacerbated impaired learning disability without affecting Aβ deposition [34], while Richard et al. demonstrated that TLR2 eliminated Aβ and subsequently attenuated cognitive impairment in an AD animal model [35]. Notably, severe memory decline induced by TLR2 deficiency could be rescued by receiving bone marrow immune cells expressing TLR2 in AD mice, implying that the presence of TLR2 in bone-marrow-derived cells is a powerful natural endogenous mechanism to restrict Aβ toxicity and prevent cognitive impairment [9,35]. Studies regarding oncology and infection revealed that TLR2 may be involved in M2 macrophages’ polarization [36,37,38]. We observed that the expression levels of TLR2 were increased in AD and were positively associated with the infiltration level of M2 macrophages and Aβ42 levels. Taken together, our results imply that sustaining Aβ accumulation in the CNS might trigger the activation of TLR2 and promote the brain infiltration of M2 macrophages to facilitate Aβ clearance.

*FCGR2A*, also known as *CD32*, encodes one member of the immunoglobulin Fc receptor family which is related to the process of phagocytosis and the clearing of immune complexes [39]. Based on datasets from both training cohorts and validation cohorts, we found that the *FCGR2A (CD32)* expression level was upregulated in AD tissue compared to tissue in HCs. FCGR2A was positively correlated to M2 macrophages’ composition and Aβ42 level, indicating a potential mechanism for the immune infiltration of M2 macrophages.

*ITGB2,* also known as *CD18*, is a protein-coding gene, and the encoded protein also performs a vital role in immune response. ITGB2 was reported to have enhanced expression on reactive microglia in AD tissue [40]. In addition, network analysis found the expression level of *ITGB2* was involved in aging and was altered in AD [41]. CYBA is the p22phox subunit of the NADPH oxidase complex, which has been suggested to be a primary component of the microbicidal oxidase system of phagocytes [42]. Upon TLR2 activation, CYBA facilitates its phagosomal trafficking to induce a burst of reactive oxygen species (ROS) and inflammatory cytokines [43]. *NCKAP1L* encodes tissue-specific transmembrane proteins, members of the HEM family, which signal downstream in response to the engagement of various immune receptors, including TLRs [44].

In general, we found that M2 macrophages’ infiltration was most positively correlated with AD, and we found that *TLR2*, *FCGR2A*, *ITGB2*, *NCKAP1L* and *CYBA* are involved in the process of infiltration, providing new and promising insights into the treatment of AD. However, there are some limitations to this work. In particular, the infiltration of M2 macrophages and the function of hub genes is not well substantiated in experiments. LM22 is constructed as a gene signature matrix containing gene information regarding 22 human hematopoietic cell phenotypes, including seven T-cell types, naïve and memory B cells, plasma cells, natural killer (NK) cells and myeloid subsets. Peripheral macrophages and microglia homologously arise from embryonic yolk sac (YS) precursors [45]. Certain gene signatures may be shared between these cell types. Though LM22 does not contain the gene signatures of microglia, astrocytes and other brain immune cells, we cannot rule out the possibility that resident microglia might develop the profile of M2 macrophages. Thus, based on the results presented here, the origin of these brain-infiltrated M2 macrophages need to be determined with further studies. However, it is not easy to distinguish activated microglia and peripheral macrophages due to their similarity in terms of expressed cell surface markers and signaling proteins. Creating chimeric mice with the transplantation of bone marrow stem cells is a valid method to identify infiltrating macrophages, which might be helpful in addressing these questions.

## 5. Conclusions

Overall, we found that the infiltration of M2 macrophages increased in AD tissue using a combination of WGCNA and the CIBERSORT algorithm. Five hub genes were identified as biomarkers which were positively correlated to the infiltration of M2 macrophages in AD and disease progression. The digital approach used here revealed a new insight that regulating M2 macrophages migrating to CNS via TLR2 might be a potential therapeutic strategy for AD. In the future, additional experiments are warranted to further investigate the specific mechanism of TLR2 in M2 macrophages’ infiltration to verify these results.

## Figures and Tables

**Figure 1 cells-11-02365-f001:**
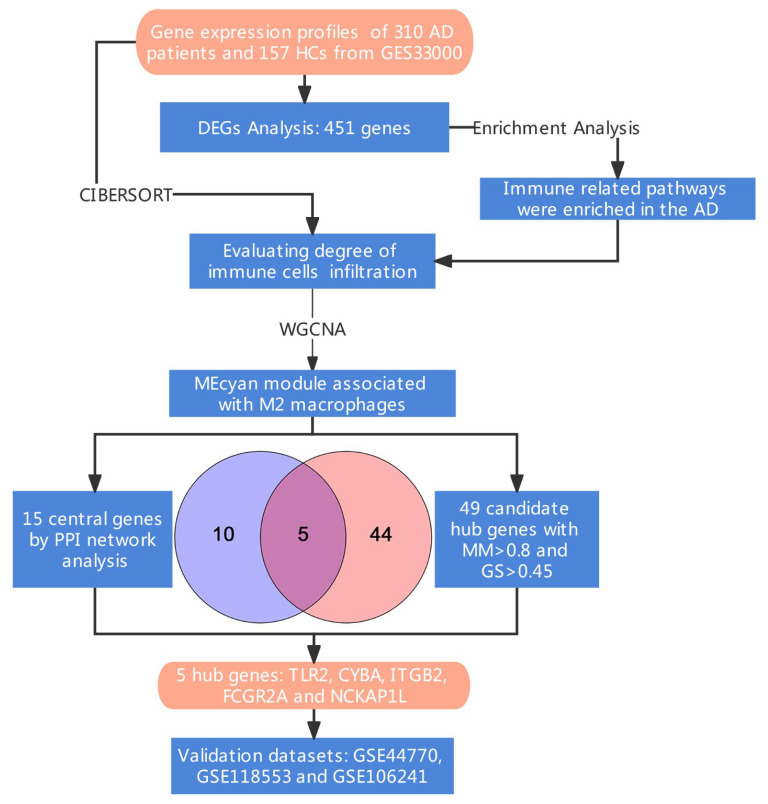
Research design flow chart.

**Figure 2 cells-11-02365-f002:**
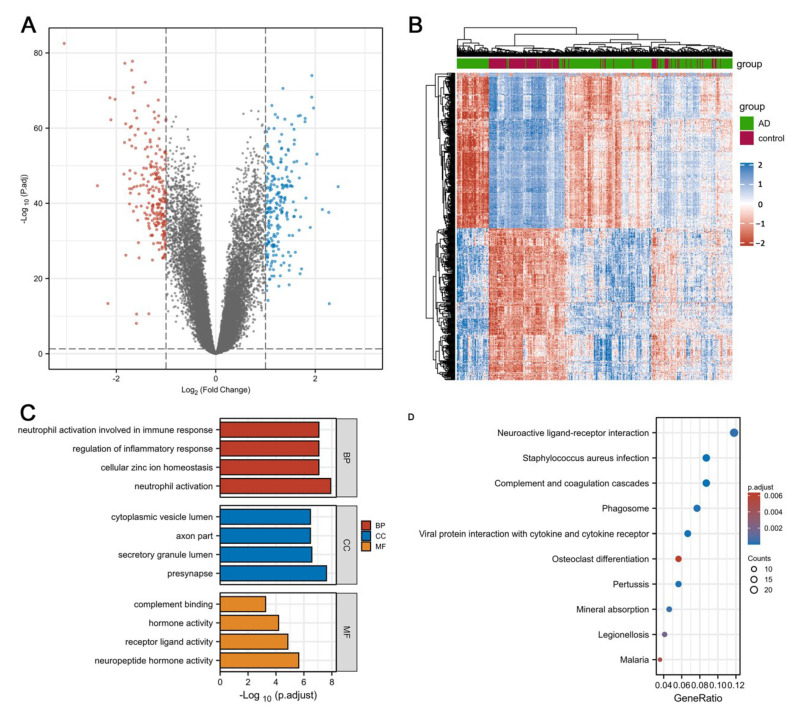
Differentially Expressed Genes and Enrichment Analysis. (**A**,**B**) Volcano map and heatmap of DEGs. (**C**,**D**) GO and KEGG enrichment analysis of DEGs.

**Figure 3 cells-11-02365-f003:**
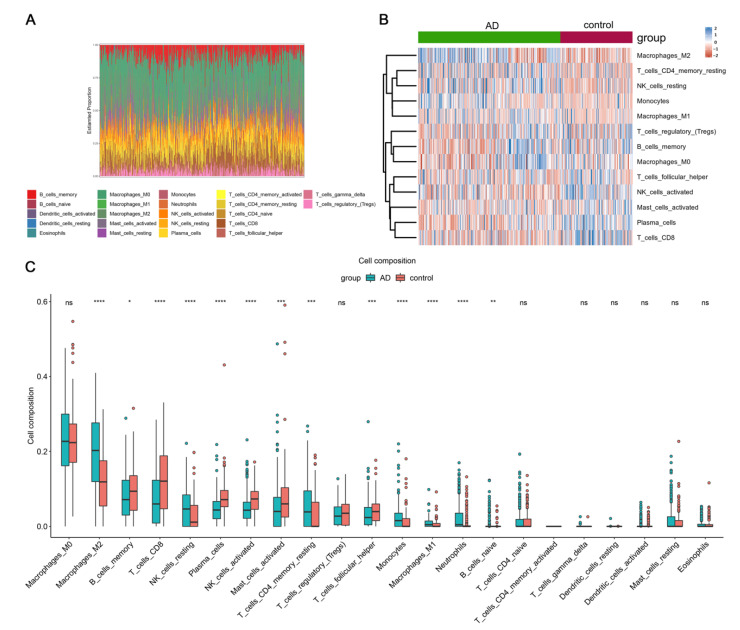
Evaluation of brain-infiltrating immune cells. (**A**) Boxplot of distribution of immune cells in each sample, the height of each color represents the percentage of such cells in the sample. (**B**,**C**) Heatmap and violin plot showing the distribution of 22 types of immune cells in AD patients and HCs. ^ns^ *p* > 0.05, * *p* <  0.05, ** *p* <  0.01, *** *p* <  0.001, **** *p* <  0.0001 compared to HCs.

**Figure 4 cells-11-02365-f004:**
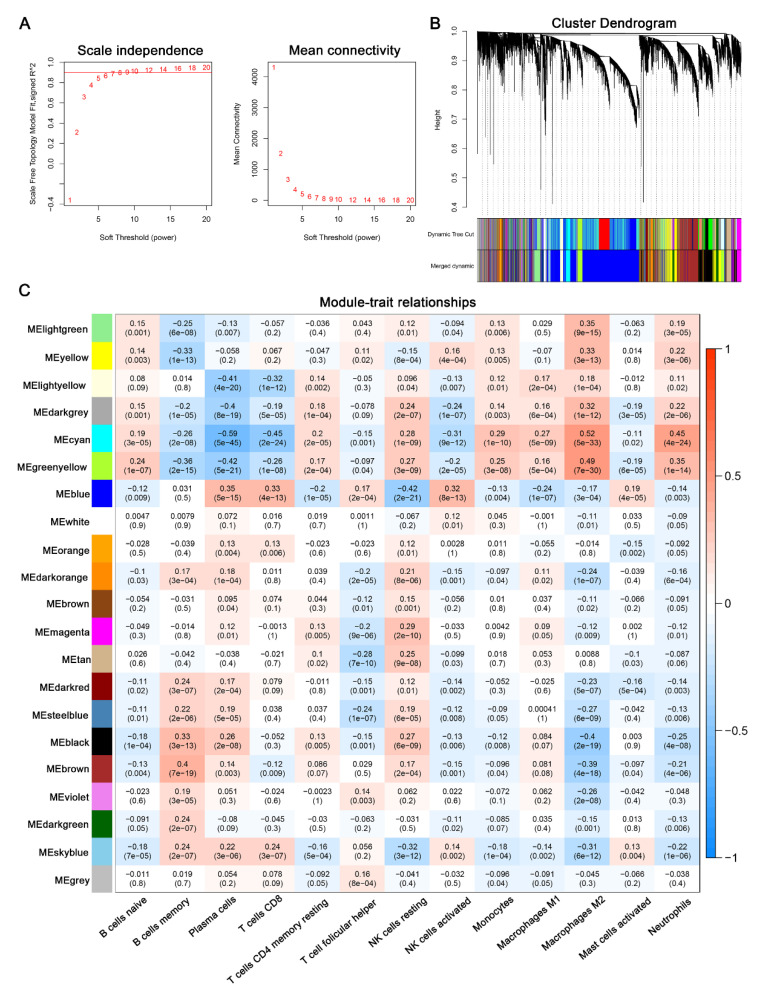
Weighted co-expression network analysis. (**A**) Analysis of the scale independence index (left) and the mean connectivity (right) for various soft-thresholding powers. (**B**) Clustering dendrogram of recognition modules. Each module was given an individual color as an identifier, including 21 different modules. (**C**) Heatmap of module–trait relationships between infiltrating immune cells and module eigengenes.

**Figure 5 cells-11-02365-f005:**
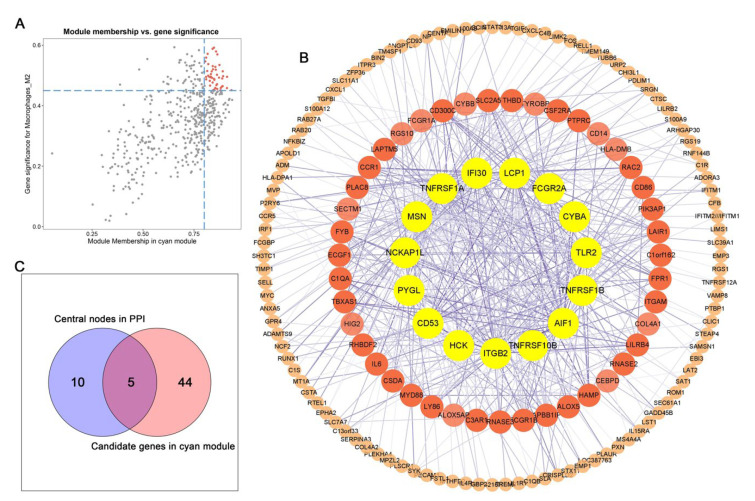
Identification of hub genes. (**A**) Scatter plot of the genes in the cyan module. Red dots indicate candidate genes of MM > 0.8 and GS > 0.45. (**B**) PPI network constructed from genes in the cyan module. Yellow nodes were selected as hub nodes (cut-offs of weight >0.15 and connectivity >15). (**C**) Venn diagram indicating five overlapped genes from hub nodes in PPI and candidate genes in cyan module.

**Figure 6 cells-11-02365-f006:**
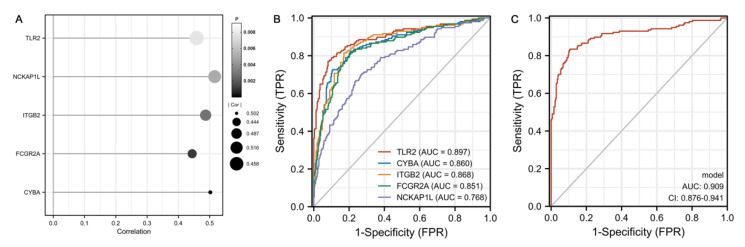
Correlation between hub genes and M2 macrophages and ROC analysis. (**A**) Lollipop diagram of the relationship between the five hub genes’ expression and M2 macrophages’ infiltration. (**B**,**C**) ROC curve analysis of the five hub genes and diagnostic model for predicting AD.

**Figure 7 cells-11-02365-f007:**
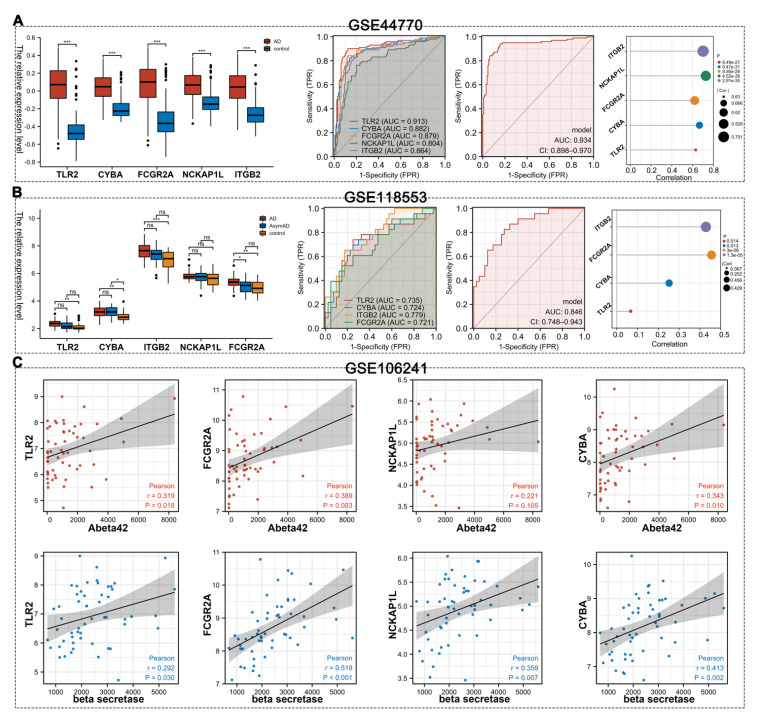
Validation of hub genes. (**A**) Validation of hub genes in the GSE44770 dataset. (**B**) Validation of hub genes in the GSE118553 dataset. (**C**) The relationship between hub genes and Aβ42 and β-secretase activity based on GSE106241. ns *p* > 0.05, * *p* < 0.05, ** *p* < 0.01, *** *p* < 0.001.

## Data Availability

The data presented in this study are openly available in NCBI Gene Expression Omnibus (GEO) datasets (https://www.ncbi.nlm.nih.gov/geo/, accessed on 16 April 2022, reference number GSE33000, GSE44770, GSE118553 and GSE106241).

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
