# Peer review of "Identification of Biomarkers Related to M2 Macrophage Infiltration in Alzheimer’s Disease"

_cells, 2022, doi:10.3390/cells11152365_

Round 1

Reviewer 1 Report

The authors found M2 macrophages was most positively correlated with AD by CIBERSORT algorithm and weighted gene co-expression network analysis (WGCNA). TLR2, FCGR2A, ITGB2, NCKAP1L and CYBA were identified as hub genes correlated with M2 macrophages infiltration in AD. Furthermore, they found the expression levels of hub genes were positively correlated with Aβ42.

This is a very interesting study, but the identification of infiltrating M2 macrophages should be better substantiated. This is mainly based on a method called Cibersort and these cells may well be resident microglia that develop of profile of M2 macrophages. This is very important issue that may change quite considerably the conclusion. The only sure method to identify infiltrating macrophages is by creating chimeric mice with transplantation of bone marrow stem cells. This is also possible in patients who had chemotherapy.

Reviewer 2 Report

Neuroinflammation is thought to contribute to the onset and progression of Alzheimer’s disease (AD). The authors intend to “verify potential mechanisms” and screen out biomarkers related to immune infiltration in AD through bioinformatics based on the gene expression datasets of AD patients and healthy controls from the Gene Expression Omnibus (GEO) database. They found that M2 macrophages were most positively correlated with AD among brain-infiltrating immune cells by use of a CIBERSORT algorithm and weighted gene co-expression network analysis (WGCNA). They also identified TLR2, FCGR2A, ITGB2, NCKAP1L and CYBA as hub genes, of which expression levels correlated with infiltration of M2 macrophages in AD. They show that expression levels of these hub genes correlate with Aβ42 levels and β-secretase activities in AD. They also show that the AUC values of these hub genes are high for AD. They conclude that their work expands our understanding of immunological mechanisms of AD and provides new insights of therapeutic strategy in AD. The authors’ findings are new and interesting. The authors should consider the following points to improve their paper.

11.        Main inflammatory cells in the brain are microglia and astrocytes. The gene expression datasets from GEO database contain expression data from microglia and astrocytes. Many of immune genes are commonly expressed in peripheral and central immune cells. Figure 2 and 3 do not contain microglia, astrocytes and other brain immune cells. How are the expression data from microglia, astrocytes, and other immune cells in the brain eliminated or removed from the analysis to identify brain-infiltrating immune cells/cell types?

22.      TLR2 and CD32 may be considered as M1 markers. Please verify these markers.

33.    There are several lines of transgenic mouse models of AD, which express mutant forms of both APP and PS1. Please specify the animal model of AD, which was used for this paper. Please indicate the number, sex and genetic background of the mice for this paper.

44.       RNA and protein samples were obtained from “prefrontal” cortex tissues of mice. This is somewhat unusual for mice due to the size and pathophysiology of the prefrontal cortex tissues in AD mouse models. Why?

55.      According to GO enrichment analysis, neutrophil activation is highly ranked (Figure 1). Neutrophils are the most abundant cell type in human blood. However, a very small fraction of neutrophils infiltrated the brain (Figure 2). Why?

66.       The authors eliminated the outlier samples for the sample dendrogram and trait heatmap (3.3.). How are the outlier samples selected?

77.      For the GSE118553 data set, NCKAP1L levels positively correlate to infiltration levels of M2 macrophages but there was no difference between the AD and HC groups in the NCKAP1L expression levels (3.5.). How so?

88.       The value of the experimental results from APP/PS1 mice is very limited. Particularly, the quality of the western blots is very poor and the data seem to be unreliable. This reviewer recommends omitting the western blots or the all animal data.

99.       Many figures (Figure 1, 2 and 4) are indiscernible. This review recommends selecting important figures for the text and placing the other figures in Supplemental Materials so that the figures are readable.

110.   In Discussion, the authors argue that Aβ accumulation in the CNS might promote brain-infiltration of M2 macrophages to facilitate Aβ clearance via TLR2. However, Aβ clearance by M2 macrophages may not be effective because the authors found increased infiltration of M2 macrophages, which correlates with increased Aβ42 levels and β-secretase activities. Thus, regulating brain infiltration of M2 macrophages via TLR2 may not be effective for AD treatment.

111.   There are places where English editing is required:

a.       Lines 17, 38, 39, 220

b.       Lines 265-267.

c.       Lines 279-282: Please remove these sentences.

d.       Lines 323, 327

e.       Lines 364-366.

Reviewer 3 Report

The manuscript by Lin et. al., entitled “Identification of biomarkers related to M2 Macrophages infiltration in Alzheimer’s Disease” where the objective is to verify potential mechanisms and screen out biomarkers related to immune infiltration in AD.

One of the paths for the detection of AD is directed to the search for biomarkers, in this sense the present work is of great importance.

I will make some notes about the work:

- Could be updated DA 2010 data to current data in introduction

- About the animals, could be informed about how they were kept in the vivarium at the time of the study.

- To be shown to the reader, the experimental design item could be designed and incremented.

- In figure 1C and 1D need to increase the font

- The limitations of the study must be clearly stated. Likewise, the highlights of the work could be described, given the importance of the findings.

Round 2

Reviewer 1 Report

The authors have addressed the limitations of their study in the discussion. However, this is a missing sections in there new text:

Based on the results presented here, the immunological mechanisms of M2 macrophages in AD need further studies to determine (their origin??). Creating chimeric mice with transplantation of bone marrow stem cells is a valid method to identify infiltrating macrophages, which might be helpful to address these questions.

Author Response

Point 1: The authors have addressed the limitations of their study in the discussion. However, this is a missing sections in there new text:

Based on the results presented here, the immunological mechanisms of M2 macrophages in AD need further studies to determine (their origin??). Creating chimeric mice with transplantation of bone marrow stem cells is a valid method to identify infiltrating macrophages, which might be helpful to address these questions.

Response 1: Thank you for your advice. We have revised this section in line 329-332.

Reviewer 2 Report

The authors properly responded to most of this reviewer’s comments and the manuscript has been revised accordingly. Unfortunately, the animal experiments were poorly carried out and the number of animals (n=3) are too small to reach a reliable conclusion. The differences in the immune system between humans and mice are well documented. This reviewer recommend omitting the animal experiments from this paper. Alternatively, the authors may perform a similar analysis on RNAseq data from animal models of AD. The authors should consider the following points to improve their paper, additionally.

1.     The quality of the western blots is poor. Multiple protein bands are observed in the western blots except CD32 and beta-actin. It is not clear which bands represent the true proteins and protein molecular weight markers are missing from the blots. There are loading lines that are not labeled.

22. Figure 3 is overlaid by Figure 4.

33. Figure 4C and Figure 7 are still unreadable.

44.      Line 335-336. Please delete “postman”.

55.       The original western images. Beta-actin is misspelled.

Author Response

The authors properly responded to most of this reviewer’s comments and the manuscript has been revised accordingly. Unfortunately, the animal experiments were poorly carried out and the number of animals (n=3) are too small to reach a reliable conclusion. The differences in the immune system between humans and mice are well documented. This reviewer recommend omitting the animal experiments from this paper. Alternatively, the authors may perform a similar analysis on RNAseq data from animal models of AD. The authors should consider the following points to improve their paper, additionally.

Point 1:  The quality of the western blots is poor. Multiple protein bands are observed in the western blots except CD32 and beta-actin. It is not clear which bands represent the true proteins and protein molecular weight markers are missing from the blots. There are loading lines that are not labeled.

Response 1: Thank you for your comments and advice. As you said, the differences in the immune system between humans and mice are well documented. In our work, we have validated the results using several GEO expression profiles of AD patients. It is no need to perform animal experiments to verify the results. So, according to your advice, we deleted the the animal experiments from this paper in the revised version.

  1. Figure 3 is overlaid by Figure 4.

Response 2: An error occurred while generating the pdf file. We are sorry for the mistake and re-submit a new pdf file.

  1. Figure 4C and Figure 7 are still unreadable.

Response 3: We have modified these figures.

  1. Line 335-336. Please delete “postman”.

Response 4: Correction has been made in the revised version (line 285).

  1. The original western images. Beta-actin is misspelled.

Response 5: The results of western blots have been deleted in the revised version.